# The Development of a Nationwide, Multicenter Electronic Database for Spinal Instrumentation Surgery in Japan: Japanese Spinal Instrumentation Society Database (JSIS-DB)

**DOI:** 10.3390/healthcare10010078

**Published:** 2021-12-31

**Authors:** Haruki Ueda, Hideyuki Arima, Tokumi Kanemura, Masao Koda, Mitsuru Yagi, Koji Yamada, Kazumasa Ueyama, Yukihiro Matsuyama, Hiroshi Taneichi

**Affiliations:** 1Department of Orthopaedic Surgery, Dokkyo Medical University, 880 Kitakobayashi, Mibu 321-0293, Japan; tane@dokkyomed.ac.jp; 2Department of Orthopaedic Surgery, Hamamatsu University School of Medicine, 1-20-1 Handayama, Higashiku, Hamamatsu City 431-3192, Japan; arihidee@gmail.com (H.A.); spine-yu@hama-med.ac.jp (Y.M.); 3Department of Orthopedic Surgery, Spine Center, Konan Kosei Hospital, 137 Omatsubara, Takayacho, Konan City 483-8704, Japan; spinesho@vmail.plala.or.jp; 4Department of Orthopaedic Surgery, University of Tsukuba, 1-1-1 Tennodai, Tsukuba City 305-8576, Japan; masaokod@gmail.com; 5Department of Orthopaedic Surgery, Keio University School of Medicine, 35 Shinanomachi, Shinjuku Ward, Tokyo 160-8582, Japan; yagiman@gmail.com; 6Nakanoshima Orthopaedics, F&F Haimu, 6-26-2, Nakanoshima, Tama-ku, Kawasaki City 214-0012, Japan; forpatients2008@gmail.com; 7Department of Orthopaedic Surgery, Aomori Jikeikai Hospital, 146-1, Chikano, Yasuta, Aomori City 038-0021, Japan; ueyamakz@yahoo.co.jp

**Keywords:** multicenter clinical database, spinal instrumentation surgery, clinical registry

## Abstract

(1) Background: Despite the number of complicated and expensive spine surgery procedures maintained by the national health insurance system in Japan, until now there has been no large-scale multicenter clinical database for this field to understand and improve healthcare expenditure and treatment outcomes. The purpose of this report is to announce the establishment and methodology of a nationwide registry system for spinal instrumentation surgeries by the Japanese Spinal Instrumentation Society (JSIS), and to report the progress over the first 1.5 years of this database’s operation. (2) Methods: The JSIS recently produced an online database with an electronic server. The collected information included patient background, surgery information, and early complications of primary and revision cases. Analysis included data from February 2018, when registration began, to August 2019. (3) Results: As of August 2019, 73 facilities have completed the required paperwork to start, and 55 facilities have registered cases. Of the total 5456 registered cases, 4852 were valid and 2511 were completed. (4) Conclusions: JSIS-DB, the nationwide web-based registry system for spinal instrumentation surgery in Japan, was launched for the purpose of research, healthcare policy regulation, and improved patient care, and its methodology and progress in the first 1.5 years are reported in this study.

## 1. Introduction

Japan’s population is aging rapidly. In 2019, the total proportions of the population aged 65 and over and 75 and over were 28.4% and 14.7%, respectively [1]. Healthy life expectancy, the period during which daily life is not hindered, has been lengthened; however, the gap between healthy life expectancy and life expectancy was estimated to be more than 8 years in males and 12 years in females in 2016. The employment rate of people aged more than 70 keeps rising over the last decade [1]. This social trend implies a growing expectation towards an active and independent lifestyle for the elderly, and the locomotive syndrome, which denotes disorders from the deterioration of locomotor organs, has been recognized as one of the key problems to tackle as part of Japanese national policy [2]. As a reflection of this trend, more people suffer from spinal diseases and undergo spine surgeries. The procedures become more complicated in relation to aging and degenerative diseases, and in an increasing number of cases spinal implants are utilized [3]. Despite this expanding trend of complicated and expensive surgical procedures maintained by the national health insurance system, there has been no nationwide clinical database of spinal instrumentation surgeries in Japan. To comprehend the current nature of spinal instrumentation surgeries at a national scale and to improve healthcare expenditure and treatment outcomes—specifically, the safety and durability of these surgeries—the Japanese Spinal Instrumentation Society (JSIS) has established a registry system for spinal instrumentation surgeries. The system started independently in February 2018 as the first generation and was modified to the second generation in October 2020 to share core patient information with the Japanese Orthopaedic Association National Registry (JOANR). 

The purpose of this report is to announce the establishment and methodology of the Japanese Spinal Instrumentation Society Database (JSIS-DB), and to report on progress during the early phase of this database.

## 2. Materials and Methods

The primary goal was to discover the required resources, safety, and durability of spinal instrumentation surgeries to treat spinal diseases. The online database with electronic server was newly developed (FAST, Inc., Spotsylvania, VA, USA). This project was produced by the JSIS, assessed by its ethics committee, and approved by its board of directors. This database was created with a subsidy for the Project for Developing a Database of Clinical Outcome approved by the health policy bureau of the Ministry of Health, Labor, and Welfare.

### 2.1. Participating Institutions

The facilities of JSIS members were eligible, and the board members and councilors of the society were strongly encouraged to register eligible cases performed in their affiliate institutions as the requisite to maintain member status. Each participating institution was required to obtain the approval from the ethics committee of its facility individually to join the project.

### 2.2. Patient Eligibility and Consent

The eligible patients/surgeries are as follows: (1) patients who underwent spinal instrumentation surgery at facilities to which society members belong; (2) inpatient surgeries; (3) spine surgeries with implants for fusion or stabilization of the spinal segment, or internal fixation for spinal fractures or spondylolisthesis; (4) revision surgeries for the index and/or equivalent level of (1)–(3). The following are excluded: (1) biopsy, (2) standalone vertebroplasty/balloon kyphoplasty, (3) standalone laminoplasty, and (4) spinal instrumentation utilized for non-spinal fractures such as pelvic fractures. 

Eligible patients were fully informed by written documents or disclosed materials available on the JSIS-DB website (http://jsisdb.org (accessed on 30 December 2021)). Patients’ consent was documented in the written consent form or medical chart, in the way that each participating facility employed. Opt-out could be also selected by invoking the patients’ right to reject or withdraw from participation.

### 2.3. Data Collection

Data managers or surgeons who completed a short e-learning course could input the data through the website secured by the assigned account and password, but surgeons needed to confirm completion of the input of correct information. The system was equipped with an auto alert function for missing components or outliers. The number of spinal instrumentation cases of a previous year in each participating facility was self-reported beforehand and utilized to estimate the expected volume of eligible cases. Occasionally, the database secretariat division sent an email to encourage the participants to input cases when the number of their registered cases was much lower than the estimate.

Personal information, including raw ID number in the hospital, was prohibited from being input into the database, and another secure identification number was required to be assigned for data management purposes in each facility, by which only the facility could identify the patient. Instead, in order to track the same individual in terms of revision surgeries even across the institutions, a hash value was created from the patient’s name, sex, and date of birth by the dedicated application and recorded. The collected information included patient background (age, height, body weight, body mass index (BMI), comorbidities, diagnosis by ICD-10, and physical status classification by American Society of Anesthesiologists (ASA-PS)), surgery information (date of the surgery, instrumented spinal levels, type of surgery by K-code, operation time, estimated blood loss, primary/revision surgery, years of surgical experience of the operators/assistants, complications within 14 days after operation, surgical procedures, and utilized implants), and additional information for revision surgeries (date, facility, diagnosis, and procedures of the primary surgery; reason/s for revision; procedures; and removed implants during revision surgery).

### 2.4. Data Retrieval and Analysis

The data from each facility was pooled in the server. While each facility had access only to its own patients’ data, the system manager in the central database office could retrieve all the data from the server in individually unidentifiable form through the web system in CSV format. The retrieved data file was opened in Microsoft Excel and analyzed. Data registration started in February 2018. This analysis included data from February 2018 to August 2019.

Statistical analysis was performed with JMP software (version 12.0, SAS institute Inc., Cary, NC, USA). Valid answers for each item from both incomplete cases and completed cases were counted. Some questions allowed multiple answers. The frequency of the answers was counted. A comparative analysis was performed by Wilcoxon rank sum test.

## 3. Results

### 3.1. Development of Our Database

As of August 2019, 73 facilities had completed the required paperwork to start, and 55 facilities (52 of 81 councilor facilities, and three additional voluntary facilities) had started inputting data. At the end of the first-generation JSIS-DB in September 2020, 91 facilities had completed the required paperwork to participate and 80 facilities had started registering cases. As of March 2021, by which time the second-generation JSIS-DB had completed the first six months of sharing core information with JOANR, 90 facilities had registered cases. The input form of the first-generation database is presented in Appendix A.

### 3.2. Initial Data

As of August 2019, a total of 5456 cases had been registered from 55 facilities. Among them, there were valid responses for 4852 cases, and the number of completed cases was 2511 (Figure 1). It is worth noting that the statistical results in the figures and tables in this study are from these valid cases and are not limited to the completed cases, and the total number of each item may vary.

The average age at time of surgery was 62 years, with bimodal distribution in teenage years and around the 70s (Figure 2). Nearly three-quarters of patients had some medical comorbidities according to the ASA-PS (Table 1). A comparison between primary and revision cases is shown in Table 2. Primary surgery tended to take longer, with more estimated blood loss, but the number of days to discharge was not significantly different compared to the revision cases. Revision surgery was more frequently performed by open surgery rather than a less invasive approach. Detailed information for the primary cases is presented in Table 3, Table 4, Table 5, Table 6 and Table 7 and Figure 3. The most frequent diagnosis for primary surgery was degenerative diseases, including spondylolisthesis and lumbar canal stenosis, explaining nearly 40% of the total, followed by spinal deformities, including idiopathic scoliosis (Table 3). The most common surgical procedures were posterior interbody fusion and posterior/posterolateral fusion, comprising more than 50% of all procedures (Table 4). In Table 5, the use of intraoperative assistive devices is reported. The prevalence of these devices, including blood salvage, neuromonitoring, and intraoperative CT/Navigation, is important for surgical safety and healthcare expenditure in our system because usage of these can be claimed for public insurance. In terms of spinal implants, screws and rods are very common, and hooks are few in number (Table 7). The reasons for the revision surgeries/additional surgeries are provided in Table 8. Implant issues and nerve symptoms are the two major reasons for revision surgeries besides planned reoperations, which include the removal of temporal internal fixators for spinal fractures and growth-friendly surgeries for early onset scoliosis.

## 4. Discussion

Currently, in the field of spine surgery, various treatment strategies have been developed, are available, and will continue developing [4]. The decision on which procedures to use is largely the responsibility of surgeons. This choice should be the preferred and familiar alternative for performing surgeons in order to minimize complications [5,6]. However, decisions should also be made based on “the conscientious, explicit, and judicious use of current best evidence”, as described by Sackett et al. [7]. In an aging country with longer life expectancy, which is managed by a national healthcare system, a safer and effective treatment choice with less medical expense must be enthusiastically explored. 

### 4.1. Database Creation

We need the information that would support the development and promotion of new safe surgical technologies, help surgeons select more effective treatment strategies with better outcomes, and that could be utilized as a resource for national healthcare policymakers to distribute public health insurance to cost-effective treatment by adjusting the surgical fee. Hence, a large-scale clinical database to record information and present an accurate view of the current Japanese situation in relation to spine surgery is essential. In the JSIS-DB, many items relating to clinical expense will be documented, for example, operative time, blood loss, and complications. These will provide us with the information required to compare various surgical procedures in terms of invasiveness and surgical risks. In addition, questions about the required resource, for example, the number of years of experience of the participating surgeons, the number of technicians/nurses, the types of implants used, and the number of days before discharge, are to be answered. This information will clarify what medical resource is required to treat the disease with the index procedures as well as the current trend of the treatment choice. As far as we could ascertain, the quality and quantity of human resources are unique items for a question about large-scale clinical databases. We believe the data from our database will help healthcare policymakers understand the effort required and difficulty of the procedures and regulate the surgical fee appropriately based on the practical costs.

Essentially, facilities with board members and councilors of the society were the intended participants, and participation in this database project was one of the requisites for applying or maintaining society councilor status. Even among these core members of the society, participation rate has not reached 100%. The cumbersome paperwork, including the application to the ethics committee and obligated e-learnings, is likely an obstacle to participation, and this complexity derives from the balance between the patients’ identity and their personal data protection. As meaningful information, particularly for revision surgeries, the individual cases have to be tracked chronologically as the same individuals. However, those individuals must not be identified personally. In the Personal Information Protection Law of our country, personal information can be utilized in an unidentifiable form if the academic research organization or group uses it for the purpose of academic research. For this reason, this registry must be conducted by the JSIS as a research organization in the form of multicenter joint research with one research plan; each facility participates as one of the joint research institutions, each with its own ethical approval for the shared research plan, and hash value is provided for each participating case to link the same individual without infringing upon his/her privacy. All the necessary documents for the procedure, including the research plan, document for the informed consent, and certificate of ethical approval by the institutional review board (IRB) at the flagship institutions, are available on the database website, with detailed information and frequently asked questions (FAQs) intending to lessen the burden of the surgeons. The database secretariat division repeatedly promoted this database and shared essential information via emails as well as on the website to encourage participation and answered any questions from participants. In this project, in addition to the ethical approval as multicenter database research by the society’s ethics committee, each participating facility was required to obtain its own ethical approval through their IRB, which makes the participating procedure more cumbersome. In some other countries, for example, the United States and certain European countries, this process has been simplified to facilitate clinical research. The United States Food and Drug Administration (FDA) makes some research exempt from IRB review, or accepts centralized IRB and joint reviews that cover all the participating facilities in multicenter clinical studies (FDA 45 CFR 46), for example, in American Academy of Orthopaedic Surgeons (AAOS) registries. Some European countries accept ethical approval by one single national ethical committee for facilities in the country to join multinational non-interventional observational clinical studies [8]. The other reasons not to participate were that the facility had stopped operating and that there were no cases to register. 

To promote data correction, non-medical data managers could start registering cases and continue to the end, including early complications within 14 days after surgery, but completing the system required a review and authentication of content by the surgeons. This registration/authentication system was intended to maintain the accuracy of the input information through the surgeons’ verification while lessening the burden on the surgeons with the help of data managers. For better data quality, we took various measures. A clear definition of each term is available in a pop-up sidebar next to the data form, and FAQs and email access to the database secretariat are available to guide a correct response (Appendix B). In addition, several auto-check functions are built into the system. For example, when the required field is blank, red alert signs on the summary page appear one step before the completion button. Significantly deviated numbers, e.g., 105 years for age or 150 kg for body weight, require confirmation of the answer before finishing. However, the persistence of obviously wrong information, for example, 10 August 1937 as operation day, likely a case of mistaking date of surgery for date of birth, cannot be prevented. Data cleansing with perseverant correction is the ideal key, but its cost and manpower are problems that need to be solved.

### 4.2. Data Interpretation

More than 5400 cases were registered, and about 2500 cases were completed for all the items in the form. Roughly 3000–4500 valid answers were obtained in each item. This relatively low completion rate among the total cases can be attributed to the authentication step of the system, which must be completed by surgeons. 

The collection of data from thousands of cases of spinal instrumentation surgeries in this short period of time in Japan is unprecedented. These data delineate the current trend of spinal instrumentation surgery in this country. The age distribution shows a two-peak pattern, with a larger spike in the elderly (Figure 2). More than 600 people over 80 years old underwent spine instrumentation surgeries, and our patient population was much older than the reports from Italy and the U.S. on spinal fusion surgery trend [9,10]. The aging of Japanese spine surgery patients has been pointed out, and our result affirms this trend (Table 9) [3,11]. Reflecting this aging, three-quarters of the patients had some systemic comorbidities and were classified as ASA-PS 2 to 4 (Table 1). The younger age peak was during teenage years. There were 319 idiopathic scoliosis cases (8.9%) of primary diagnosis, and the characteristic age distribution of this disease likely explains the peak. Many of the participating facilities are regional spine centers to which scoliosis cases are referred, and this may also increase the density of surgical cases for this disease population in this database. The two most frequent diagnoses were spondylolisthesis and spinal canal stenosis (Table 3), and the two most frequent procedures for primary surgery were posterior interbody fusion and posterior/posterolateral fusion (Table 4). One-third of the total cases were either of these two diagnoses for degenerative spine disease. Upper instrumented vertebra (UIV) and lower instrumented vertebra (LIV) are common in the lumbar spine (Figure 3), and these suggest that many of the cases are degenerative lumbar spine treated with posterior procedures with short fusion levels. Kyphosis and scoliosis make up about 25% of the total cases. Traumas, including fractures and dislocations, explain less than 10% of the total (Table 3).

In most of the cases, a certain kind of screw was used, such as pedicle screw, cortical bone trajectory, lateral mass screw, or vertebral screw. Hooks and tapes/wires are not as common anymore. Cervical artificial discs are not approved for general use at the time of this analysis, and these cases are from some trial facilities (Table 7).

Now that the multi-center database is established and has started accumulating a large amount of data regarding spinal implant surgeries, more than 90 operating facilities have joined, and the number will increase. Geographically, these facilities cover the whole area of Japan, and this database will capture nationwide information. This report only contains the global trend of the first-year data, but more detailed information, for example, trends in age groups, geographic areas, or surgical approaches, could be extracted from this database. 

One of the points to solve is the data quality and quantity. Ideally, all the surgeons thoroughly recognize the importance and the meaning of the data correction and are highly motivated to cooperate towards the better. Useful and meaningful information as the product from this database will motivate them. The requirement to stay as the councilor seems to be an effective means with which to increase the number of participating facilities. The Japanese Orthopaedic Association National Registry (JOANR) was launched in April 2020 as the surgical case registry for members accepting all kinds of orthopedic operative procedures, and it functions as the surgical case list of each surgeon to be assessed or the application and maintenance of accredited specialist status by the JOA. JSIS-DB has started sharing basic patient information with JOANR, which avoids inputting the same data twice. Concretely, after filling all the basic information in the JOANR site, the link to the JSIS-DB becomes available for spinal instrumentation cases. Through this link, basic information of the same patient is already filled in the JSIS-DB, and only additional fields should be filled out for more specific information of the spinal instrumentation surgery. This sharing is expected to collect not only more cases but also more facilities for the JSIS-DB, from those who are currently not participants of the JSIS.

In summary, JSIS-DB has been launched, and the data from the first 1.5 years have been analyzed. These data describe the trend of spinal instrumentation surgeries. The number of participating facilities is increasing, and more cases are expected to be registered. By accumulating more cases while maintaining the high quality of the data, JSIS-DB will provide powerful and useful information to improve healthcare policy and patient outcomes for spine surgery in Japan.

## 5. Conclusions

JSIS-DB, a nationwide web-based registry system for spinal instrumentation surgery in Japan, was established for the purpose of research, healthcare policy regulation, and improved patient care, and its methodology and progress in the first 1.5 years was reported in this article. 

## Figures and Tables

**Figure 1 healthcare-10-00078-f001:**
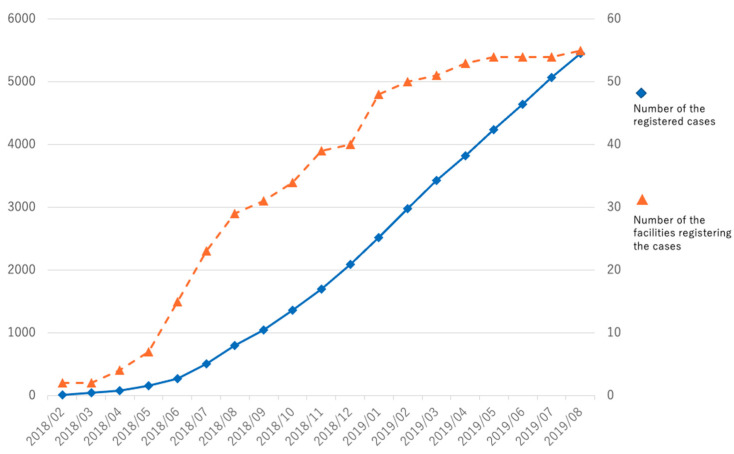
Cumulative number of registered cases and registering facilities. ◆ Number of registered cases on the left vertical axis. ▲ Number of facilities registering cases on the right vertical axis.

**Figure 2 healthcare-10-00078-f002:**
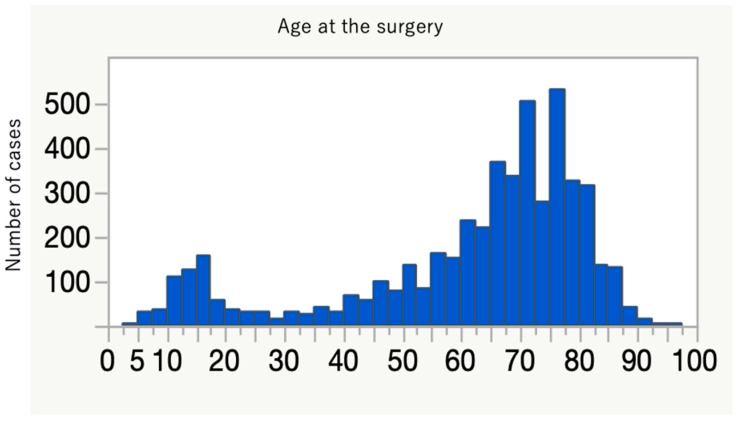
Distribution of patient age at time of surgery, showing bimodal peaks for young and old populations.

**Figure 3 healthcare-10-00078-f003:**
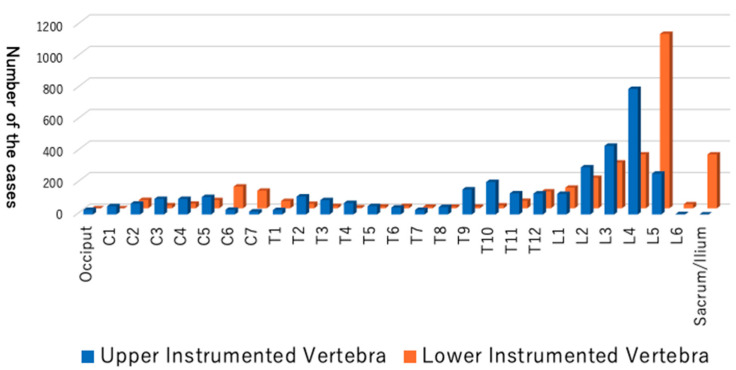
Distribution of the upper instrumented vertebra (UIV) and the lower instrumented vertebra (LIV).

**Table 1 healthcare-10-00078-t001:** Characteristics of registered cases.

		Number	%
Sex	male	2218	46
	female	2634	54
ASA-PS *	ASA 1	1243	28
	ASA 2	2553	58
	ASA 3	546	13
	ASA 4	30	1
Elective/Emergency	Elective	4157	95
	Emergency	215	5
Primary/Revision	Primary	4373	84
	Revision	691	15

* ASA-PS: physical status classification by the American Society of Anesthesiologists.

**Table 2 healthcare-10-00078-t002:** Comparison between the primary and revision cases (Wilcoxon rank sum test).

		Primary Surgery	Revision Surgery	*p*
Number of cases		3660	682	
Operative time (min) (SD *)		207 (111)	191 (122)	<0.0001
Estimated blood loss (g) (SD)		350 (498)	344 (542)	0.008
Days to discharge (home/hospitals/facilities)		24 (20)	20 (10)	0.64
Procedure (%)	Open	79	92	
	Mini-open	13	6	
	MIS/Percutaneous	8	1	
Top 3 early complications		Incidental Durotomy	Incidental Durotomy	
		Nerve root injury	Nerve root injury	
		Implant breach/misplacement	Implant breakage/dislocation	

* SD = Standard deviation.

**Table 3 healthcare-10-00078-t003:** Diagnosis for primary surgery.

Diagnosis	*n*	% *
Kyphosis	238	6.7
Infantile idiopathic scoliosis	12	0.3
Idiopathic scoliosis	305	8.6
Other scoliosis	379	10.7
Spinal canal stenosis	624	17.6
Spondylolysis	68	1.9
Spondylolisthesis	710	20
Ankylosing spondylitis	1	0.03
Ankylosing spinal hyperostosis	8	0.2
Ossification of Posterior Longitudinal Ligament (OPLL)	108	3
Myelopathy	170	4.8
Radiculopathy	68	1.9
Spondylitis, discitis	58	1.6
Spinal caries, vertebral tuberculosis, and other infectious spinal disorders	7	0.2
Cervical disc herniation	61	1.7
Disc herniation in other parts	162	4.6
Osteoporotic fractures	44	1.2
Fracture in the cervical spine	55	1.6
Dislocation in the cervical spine	34	1
Cervical spinal cord injury	20	0.6
Fracture in the thoracic spine	76	2.1
Dislocation in the thoracic spine	5	0.1
Thoracic spinal cord injury	2	0.06
Fracture in the lumbar spine	129	3.6
Dislocation in the lumbar spine	4	0.1
Nerve injury in the lumbar spine	0	0
Malignant tumor in the spine	56	1.6
Metastatic tumor in the spine	34	1
Epidural abscess	2	0.06

* % in the valid 3546 cases.

**Table 4 healthcare-10-00078-t004:** Main surgical procedures for primary surgery (single answer allowed).

Main Surgical Procedures	*n*	% *
Posterior interbody fusion	1349	38.5
Posterior fusion, posterolateral fusion	863	24.6
Fusion for scoliosis	464	13.2
Anterior interbody fusion	409	11.7
Anteroposterior fusion	286	8.1
Vertebral osteotomies	36	1
Rod instrumentation for growth-friendly surgery	34	1
Total spondylectomy	13	0.4
Endoscopic spinal fusion	7	0.2
Others	43	1.2

* Among valid 3504 cases.

**Table 5 healthcare-10-00078-t005:** Additional intraoperative assists.

Additional Intraoperative Assists	*n*
Intra- or postoperative blood salvage	1050
Neuromonitoring	2131
Imaging support (CT *, Navigation, etc.)	759

* CT = Computed tomography.

**Table 6 healthcare-10-00078-t006:** Early complications within 2 weeks after primary surgery (multiple answers allowed).

Complications	*n*
Incidental durotomy	110
Nerve root injury	68
Hematoma	42
Implant breach/misplacement	40
Psychiatric disorder	35
Respiratory complications	23
Spinal cord injury	20
Implant breakage/dislocation	16
Dysphagia, airway obstruction	16
Cardiac/circulatory complications	16
Gastrointestinal complication	15
Pulmonary embolism/deep vein thrombosis	11
Cauda equina injury	6
Hemothorax, pneumothorax, and chylothorax	5
Death	4
Major vascular injury	3
Operated on wrong spinal level	2
Head injury	1
Bowel injury	0
Ureter injury	0
Others	98
None	2972

**Table 7 healthcare-10-00078-t007:** Implant used (multiple answers allowed).

Implants	*n*
Screw (pedicle, CBT *)	2966
Screw (others)	230
Hook	162
Sublaminar tape, wire	265
Rod	2851
Transverse fixator	930
Plate	154
Intervertebral cage (posterior)	1407
Intervertebral cage (anterior)	552
Cage for vertebral replacement	97
Laminar spacer (artificial bone)	28
Laminar spacer (screw)	10
Cervical artificial disc	18
None	17
Others	40

* CBT = Cortical bone trajectory.

**Table 8 healthcare-10-00078-t008:** Reasons for revision surgeries/additional surgeries.

Reasons	Number	% *
As planned	96	18
Hematoma	3	0.6
Surgical site infection	18	3.4
Nerve root injury	53	10
Spinal cord injury	31	5.8
Cauda equina symptom	36	6.8
Implant breakage/dislocation	77	14.4
Implant breach/misplacement	18	3.4
Implant loosening	29	5.5
Vertebral fracture	28	5.3
Non-union/pseudoarthrosis	22	4.2
Malalignment	37	7
Others	84	15.8

* Rate in the 532 valid answers.

**Table 9 healthcare-10-00078-t009:** Summary of other database studies on spine surgery.

Author	Objectives	Method	Sample Year	Number of Institutions	Number of Cases	Mean Age
Goz [9]	Spinal fusion	Nationwide Inpatient Sample Database in the US	2001–2010	NA	3,552,873	49 (2001)–54 (2010)
Cortesi [10]	Spinal fusion	Healthcare administrative database	2001–2010	NA	17,772	55
Imajo [3]	Decompression, fusion	Computerized questionnaire	2011	209	31,380	59.3
Kobayashi [11]	Decompression, fusion	Database in their group hospitals	2004–2015	14	45,831	55 (2004)–64 (2015)
Ueda (this report)	Spinal fusion	Multicenter online database	2018–2019	73	5456	62

## Data Availability

The data presented in this study are available on request from the corresponding author. The data are not publicly available due to the protection of privacy of participants.

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
