# Peer review of "The Development of a Nationwide, Multicenter Electronic Database for Spinal Instrumentation Surgery in Japan: Japanese Spinal Instrumentation Society Database (JSIS-DB)"

_healthcare, 2021, doi:10.3390/healthcare10010078_

Round 1

Reviewer 1 Report

I like the study in general as its results could provide possibilities for applications in many fields, aiming at a good recovery of patients. Nevertheless, I have some concerns as following:

  • In the abstract part, the authors firstly mentioned that they reported the first year of this database, which was 1.5 years actually and should be more accurate.
  • Among all tables, table 2 was mentioned firstly in the main manuscript and should be organized orderly.
  • Among the tables, the total number of the registered cases was different, which should be explained in detail.
  • The authors should explain more clearly about how the data were extracted as each facility can access only to its own patients’ data. The system manager can extract the whole data but ID number in each hospital through the web system, but is the extracted data available to all facilities?

Reviewer 2 Report

This is a well presented manuscript on the development of a nation-wide registry on spinal instrumentation in Japan. Mainly baseline characteristics and surgical data have been collected. A notable number of patient records has already been gathered. Unfortunately, the data is preliminary and information on the functional status before and after surgery has not been included. Therefore, at this point, their significance is questionable.

Reviewer 3 Report

This database is very important especially for today's aging societies all over the world. She is exemplary, thought-out and prepared. Enables a thorough analysis of spine surgery procedures. Thanks to it, you can analyze the direct costs - material costs - the cost of implants. But also the most important intangible experience of the team performing a given procedure.

The limitations of this database are the high level of necessary approvals - each center that will join in the creation of this database must have its own bioethical consent and verification of the entered data by the operator. It probably results from local law and the young, several-year-old concept of this base.

It is important that this database has operated for several years in correlation with a large - several thousand number of registered medical cases, which will enable a good statistical analysis. And the following years promise even more material for medical analysis.

This paper presents the appropriate criteria for the inclusion and exclusion of patient qualifications.

Correct and well-planned discussion is a strong point of this work.

The fact of a small number of references is also noticeable, especially from the last few years, but this may indicate a new and needed idea and concept of this Database.

Oh, and the appendix should be translated into English, unless this is the intention of the authors.
